# DANCE-ST: WHY TRUSTWORTHY AI NEEDS CONSTRAINT GUIDANCE, NOT CONSTRAINT PENALTIES

## ABSTRACT

Neural networks achieve high accuracy in spatiotemporal prediction but often violate physical constraints, creating a fundamental accuracy-safety dilemma. We introduce DANCE-ST, a constraint-guided learning framework that resolves this trade-off by treating physical laws not as adversarial penalties, but as collaborative information sources that actively guide learning. Our core contribution is a novel three-phase architecture that (1) identifies critical system components by diffusing state-dependent "constraint potentials" through a knowledge graph, (2) intelligently fuses neural and physics-based predictions with provable error bounds for asynchronous sensors, and (3) projects predictions onto the constraint-satisfying space with guaranteed linear convergence. This architecture is orchestrated by a fault-tolerant multi-agent system for robust deployment. Experiments on industrial datasets demonstrate 97.2% constraint satisfaction while achieving state-of-the-art accuracy and the fastest inference time (38.4s) among constraint-aware methods. Critically, DANCE-ST delivers superior, verifiable interpretability (4.6/5 vs 3.8/5). By design, it provides explainable insights into which system components drive constraint violations, directly addressing the transparency requirements of emerging safety regulations (e.g., EU AI Act, FDA AI guidelines) in a way black-box enforcement cannot. Our work establishes constraint-guided learning as a foundational paradigm for trustworthy AI, demonstrating that the accuracy-safety trade-off is a false dilemma when constraints become collaborative guides.

## 1 INTRODUCTION

Safety-critical applications require neural networks that respect physical laws, creating challenges across domains from autonomous systems (Amodei et al., 2016) to medical AI (Adebayo et al., 2018) and scientific computing (Karniadakis et al., 2021). In industrial systems, a turbine blade prediction of 1,250°C might be statistically reasonable but physically impossible if it exceeds the material's temperature limit of 1,200°C. Traditional approaches have faced limitations: physics-informed neural networks may ignore physics when it conflicts with data (Krishnapriyan et al., 2021), while some hard-constraint methods like DC3 can be computationally expensive for large systems (Donti et al., 2021). Recent advances, particularly HardNet (Min & Azizan, 2024), have made significant progress on constraint satisfaction with competitive accuracy, demonstrating that the accuracy-constraint trade-off can be addressed through differentiable projection layers.

**Our approach:** We introduce **DANCE-ST** (Distributed Agent Network for Constraint-Enabled Spatiotemporal prediction), which emphasizes interpretable constraint-guided learning. Rather than viewing constraints as external penalties, we demonstrate that they contain information that can guide neural networks toward better solutions while providing explainable insights into which system components drive constraint violations. This shift from constraint penalty to constraint guidance transforms physics from an external validator into an intelligent guide that reveals critical system components.

DANCE-ST addresses the challenge through three coordinated phases that transform constraints from penalties into guides (detailed in Section 4). The key contribution is providing interpretable constraint satisfaction with operational advantages for industrial deployment. The dual challenges of computational scalability and operational robustness are addressed through a two-pronged strategy. Scalability is tackled algorithmically: our relevance selection mechanism (Phase I) dynamically

identifies and focuses computation on a small, critical subgraph, addressing the scaling limitations of monolithic methods that struggle with large systems (e.g., DC3 requires $> 1000$ iterations (Donti et al., 2021); HardNet struggles beyond 1000 variables (Min & Azizan, 2024)). Separately, operational robustness is achieved through a fault-tolerant multi-agent system. This design choice is critical for deployment, enabling 2.3s recovery from agent failures versus 18.7s for a complete system restart (see Section 5), a crucial capability where downtime costs $100K - 10$ \$M per incident (Leveson, 2011).

## 1.1 TECHNICAL CONTRIBUTIONS

We present three algorithmic contributions that enable interpretable constraint satisfaction:

**(1) Dynamic Relevance Probing via Constraint-Potential Diffusion:** We introduce a training-free algorithm that identifies critical system components by propagating state-dependent "constraint potentials" through the system's knowledge graph. This provides explainable insights into systemic risks by considering both local proximity to failure and propagated influence, replacing static graph metrics with a dynamic, physics-informed approach.

**(2) Temporal-Robust Neurosymbolic Fusion:** Our method fuses neural (10Hz) and symbolic (1Hz) predictions despite asynchronous sensors, with proven error bounds that grow linearly with temporal delay, maintaining robustness under realistic sensor misalignment.

**(3) Structure-Exploiting Constraint Projection:** We exploit the monotonicity properties of physical constraints via adapted Douglas-Rachford projection, achieving convergence in approximately 135 iterations compared to generic methods.

DANCE-ST achieves 97.2% constraint satisfaction with competitive accuracy and superior interpretability (4.6/5 explainability score). We validate performance across aerospace turbine monitoring, mechanical bearing prognostics, and medical patient monitoring, demonstrating the practical value of interpretable constraint satisfaction. The primary contribution is establishing constraint-guided learning as an interpretable approach to physics-informed neural networks, providing practitioners with explainable insights into constraint-critical system components while maintaining competitive performance in the growing landscape of constraint satisfaction methods.

## 2 RELATED WORK

The challenge of achieving both high accuracy and guaranteed constraint satisfaction has driven significant research across machine learning. Table 1 compares existing approaches across multiple dimensions.

Table 1: Comparison of existing approaches for constrained spatiotemporal prediction. ✓ = strong, △ = partial, ✗ = weak/absent.

| Approach | Accuracy | Constraint Satisfaction | Scalability | Real-time Capable | Fault Tolerance | Interpret. | Key Limitation |
|---|---|---|---|---|---|---|---|
| *Recent ML Methods* | | | | | | | |
| ClimODE (Müller et al., 2024) | ✓ | △ | ✓ | ✓ | ✗ | △ | Soft constraints only |
| ConFIG (Liu et al., 2025) | △ | △ | ✓ | ✓ | ✗ | △ | Training conflicts remain |
| *Traditional Neural Methods* | | | | | | | |
| LSTM/Transformer (Vaswani et al., 2017) | ✓ | ✗ | ✓ | ✓ | ✗ | ✗ | $>30\%$ violations |
| CNN-based (Bai et al., 2018) | ✓ | ✗ | ✓ | △ | ✗ | ✗ | No physics awareness |
| *Physics-Informed Approaches* | | | | | | | |
| PINN (soft) (Raissi et al., 2019) | △ | △ | ✓ | ✓ | ✗ | △ | Ignores physics when conflicting |
| Enhanced PINNs (Chalapathi et al., 2024) | △ | △ | △ | △ | ✗ | △ | Mixture-of-experts overhead |
| *Hard Constraint Methods* | | | | | | | |
| OptNet (Amos & Kolter, 2017) | ✗ | ✓ | ✗ | ✗ | ✗ | ✗ | Requires QP reformulation |
| DC3 (Donti et al., 2021) | △ | ✓ | ✗ | ✗ | ✗ | △ | High iteration counts |
| HardNet (Min & Azizan, 2024) | ✓ | ✓ | △ | ✓ | ✗ | △ | Limited interpretability |
| **DANCE-ST (Ours)** | ✓ (Competitive) | ✓ (97.2%) | ✓ (Multi-agent) | ✓ (38.4s) | ✓ (2.3s recovery) | ✓ (4.6/5) | *Setup cost only* (120 person-hours) |

**a. Physics-informed neural networks:** PINNs integrate domain knowledge into learning, but existing approaches have limitations. **Soft constraint methods** embed physics as penalty terms $\mathcal{L} = \mathcal{L}_{\text{data}} + \lambda \mathcal{L}_{\text{physics}}$, but optimizers may prioritize data fitting over constraint satisfaction. Recent work by Tang et al. (2024) addresses this through adversarial adaptive sampling, while Liu et al. (2025) proposes ConFIG to resolve gradient conflicts. However, these approaches cannot guarantee constraint

satisfaction, necessary for safety-critical systems. **Hard constraint methods** have emerged as viable alternatives. The HardNet framework (Min & Azizan, 2024) provides theoretical guarantees with competitive performance through differentiable projection layers, but offers limited interpretability into which system components drive constraint violations. Chalapathi et al. (2024) introduces mixture-of-experts for scaling, while Le Boudec et al. (2025) develops neural solvers to enhance physics-informed methods. **Spatiotemporal physics modeling** extends these ideas to time-dependent systems. Müller et al. (2024) demonstrates physics-informed neural ODEs for climate forecasting, achieving strong results by respecting conservation laws. Zheng et al. (2025) shows that memory mechanisms improve time-dependent PDE modeling. Our work extends these concepts to distributed multi-agent architectures with fault tolerance and interpretability guarantees.

**b. Constraint Satisfaction and Optimization:** Optimization-based approaches like OptNet (Amos & Kolter, 2017) and DC3 (Donti et al., 2021) guarantee constraint satisfaction but face computational challenges for large-scale real-time applications. Dangel et al. (2024) addresses computational bottlenecks through Kronecker-factored approximations, achieving speedups while maintaining guarantees. Recent advances demonstrate multiple viable approaches to constraint satisfaction. HardNet achieves high constraint satisfaction through elegant mathematical formulation, while methods like ConFIG address training dynamics. However, these approaches treat constraint satisfaction as a final enforcement step rather than leveraging constraints as information sources throughout the learning process.

**c. Multi-Agent Systems and Uncertainty Quantification** Multi-agent architectures enable robust deployment via federated learning (Weber et al., 2024) and distributed coordination. Emerging A2A (Google & Partners, 2025) and MCP (Anthropic, 2024) standards enable interoperable systems for safety-critical applications. However, their application to physics-informed neural networks with fault tolerance guarantees remains underexplored.

**Neurosymbolic integration** attempts to bridge neural learning with symbolic reasoning (Hamilton et al., 2022), but typically handles asynchronous sensor fusion as an engineering challenge rather than providing theoretical guarantees for temporal robustness.

**Gap in Constraint-Guided Learning:** While recent progress has addressed the accuracy-constraint trade-off through various mechanisms, most approaches treat constraints as external enforcement rather than as information sources that can guide learning throughout the prediction pipeline. DANCE-ST introduces constraint-guided learning, where domain knowledge becomes an active participant in relevance selection, fusion, and projection, while providing interpretability into which system components drive constraint violations.

**Technical Contributions.** Table 2 details how DANCE-ST's algorithmic contributions address limitations in existing approaches.

Table 2: Technical innovations distinguishing DANCE-ST from existing approaches.

| Component | Existing Limitation | DANCE-ST Innovation | Contribution |
|---|---|---|---|
| Relevance Selection | Static graph metrics or fixed attention patterns | State-dependent relevance via constraint-potential diffusion | Real-time, training-free identification of critical components |
| Constraint Integration | HardNet: Post-hoc projection; ConFIG: Conflict resolution | Constraint-guided learning throughout three-phase pipeline | Proactive constraint-informed processing |
| Temporal Robustness | Ad-hoc handling of asynchronous sensors | Theoretical bounds for neurosymbolic fusion under temporal delays | Guaranteed robustness with error bounds |
| Fault Tolerance | Single-point-of-failure in monolithic systems | Dual-protocol (A2A+MCP) multi-agent architecture | 2.3s recovery vs complete restart |

**Positioning:** DANCE-ST contributes to the growing landscape of constraint satisfaction methods by introducing a constraint-guided learning framework that emphasizes interpretability and fault tolerance. While methods like HardNet excel at mathematical elegance and constraint satisfaction, DANCE-ST provides strengths in explainability and operational robustness for industrial deployment.

## 3 PRELIMINARIES

We establish the mathematical foundations for constraint-guided learning in spatiotemporal systems.

### 3.1 PROBLEM FORMALIZATION

Consider predicting temperature across a turbine blade. A neural network might predict 1,250°C based on data patterns, statistically reasonable but physically impossible if the material limit is 1,200°C. This exemplifies our core challenge: learning predictor $f : \mathcal{S} \times \mathcal{T} \to \mathbb{R}$ over spatial domain $\mathcal{S}$ and temporal domain $\mathcal{T}$ that minimizes prediction error $\|f - f^*\|_{L^2(\mathcal{S} \times \mathcal{T})}$ while satisfying physical constraints $f \in \mathcal{C}$.

### 3.2 DUAL PREDICTION FRAMEWORK

Real-world systems provide two complementary information sources, each with distinct strengths:

**Neural Predictor** $f_n(s, t)$ learns complex patterns from sensor data. While highly accurate for in-distribution predictions, it lacks awareness of physical laws and may generate impossible outputs. We quantify its confidence using ensemble-based uncertainty estimation to compute $\sigma_n^2(s, t)$ (Lakshminarayanan et al., 2017).

**Symbolic Predictor** $f_s(s, t)$ encodes domain knowledge through physics equations (e.g., heat transfer, conservation laws). While guaranteed to respect physical constraints, it may use simplified models or imprecise parameters. Its uncertainty $\sigma_s^2(s, t)$ reflects both parameter uncertainty and modeling approximations, estimated through residual analysis and domain expert calibration.

**System Structure** is represented as graph $G = (V, E)$ where vertices $V$ denote system components (sensors, actuators, physical regions) and edges $E$ encode constraint dependencies (e.g., "temperature at node A affects heat flow to node B").

### 3.3 KEY MATHEMATICAL PROPERTIES

Three properties make constraint-guided learning tractable:

**A1: Error Decorrelation.** Neural and symbolic predictors make different types of mistakes. When a neural network fails, it's often because it hasn't seen similar data before (e.g., predicting turbine behavior at 200% load when trained only up to 150%). When a physics model fails, it's mainly because it uses simplified equations (e.g., assuming perfect heat transfer). These complementary failure modes mean their errors rarely coincide (mathematically), $\rho = \mathrm{corr}(f_n - f^*, f_s - f^*) < 0.35$. This low correlation makes fusion effective: when one predictor struggles, the other often compensates (see Appendix L for detailed robustness analysis under assumption violations).

**A2: Well-Conditioned Constraint Geometry.** Physical constraints create well-behaved optimization landscapes with strong convexity parameter $\mu > 0.03$, avoiding local minima that plague general nonconvex optimization (Appendix Theorem A.1) (Boyd & Vandenberghe, 2004).

**A3: Convex Constraint Structure.** Most physical constraints (temperature bounds, conservation laws) form convex sets. Non-convex cases use convex relaxation with bounded approximation error $\epsilon_{\mathrm{relax}}$ (see Appendix D for formal constraint definitions and examples) (Ben-Tal et al., 2009).

These properties enable our three-phase approach: A1 motivates Phase II (fusion), while A2–A3 enable Phase III (projection). We empirically validate them in Section 5.

## 4 DANCE-ST METHODOLOGY

Our methodology treats physical constraints not as penalties, but as an active source of information to guide learning. This principle is implemented through a three-phase pipeline that transforms the accuracy-safety trade-off into a synergistic relationship. The pipeline first identifies the system's most vulnerable components, then fuses predictions for that critical area, and finally guarantees the output is physically sound.

## 4.1 Phase I: Dynamic Relevance Selection via Constraint-Potential Diffusion

**Intuition:** Static graph metrics are insufficient for safety-critical systems because a component's relevance is not fixed, it is highly state-dependent. A component's importance can change in seconds, whether it's the strain on a robotic joint, the voltage on a power grid, or the **pressure** in a reactor. This phase, therefore, answers the question: "Which components matter most *right now*?" To do this, we introduce Constraint-Potential Diffusion, a training-free algorithm that pinpoints these dynamically critical components by combining the system's immediate physical state with its underlying network structure.

**Our Insight:** Relevance is Both Local and Systemic. Our key insight is that a component's relevance has two ingredients: (1) **Local Risk:** Its own proximity to violating a constraint. (2) **Systemic Influence:** Its connection to *other* at-risk components. Our two-step algorithm is designed to explicitly capture both aspects.

**The Solution: A Two-Step, Physics-Informed Algorithm   Step 1: Quantifying Local Risk.**
We first calculate a *local constraint potential* $\Phi(v, x)$: an "urgency score" that skyrockets as a component's state approaches its safety limit. For constraints $g_{v,j}(x) \leq 0$, this is:

$$\Phi(v, x) = \sum_j \frac{1}{\epsilon_{\text{pot}} - g_{v,j}(x)} \quad \text{for all } j \text{ where } g_{v,j}(x) < 0 \tag{1}$$

where $\Phi(v, x)$ is the potential for a component node $v$ given the system state $x$, $g_{v,j}(x) \leq 0$ is the $j$-th physical constraint function for that node, and $\epsilon_{\text{pot}}$ is a small constant for numerical stability.

**Step 2: Propagating Systemic Risk.** Local risk is insufficient; a stable part may be influenced by a neighbor approaching its failure threshold. To capture this, we diffuse these potential scores through the knowledge graph $G$ via an iterative update:

$$\mathbf{\Lambda}^{(t+1)}(x) = (1 - \alpha)\mathbf{\Phi}(x) + \alpha \mathbf{W}^T \mathbf{\Lambda}^{(t)}(x) \quad \text{with} \quad \mathbf{\Lambda}^{(0)}(x) = \mathbf{\Phi}(x) \tag{2}$$

Here, $\mathbf{\Lambda}^{(t)}$ is the vector of relevance scores at iteration $t$, $\mathbf{\Phi}$ is the vector of initial local potentials from Step 1, $\alpha$ is a damping factor ($0 < \alpha < 1$) that balances local versus propagated influence, and $\mathbf{W}$ is the symmetrically normalized adjacency matrix of the graph $G$. For instance, if a sensor on a turbine blade tip shows a temperature approaching its material limit (creating a high local potential $\Phi$), this risk propagates through the blade's thermal model. This raises the final relevance score $\mathbf{\Lambda}$ for sensors in the cooler blade root, flagging them as important even though their local temperatures are stable. After a few iterations, the resulting scores holistically capture both local and systemic risk.

**Output of Phase I:** The top-$k$ nodes with the highest relevance scores form a constraint-critical subgraph, $G'$, focusing all subsequent computation where it matters most.

## 4.2 Phase II: Neurosymbolic Fusion with Uncertainty Quantification

**Intuition:** Having identified relevant components, we now combine neural and symbolic predictions. The key insight: weight each predictor by its confidence, giving more weight to the physics model near constraint boundaries where violations are costly.

**Fusion Strategy:** The integrated prediction weighs each model by its inverse variance:

$$f_{\text{int}}(s, t) = \Omega(s, t)f_n(s, t) + (1 - \Omega(s, t))f_s(s, t) \tag{3}$$

The optimal weight balances uncertainties while prioritizing constraint awareness:

$$\Omega(s, t) = \frac{\sigma_s^2(s, t)}{\sigma_n^2(s, t) + \sigma_s^2(s, t)} \cdot \psi_{\text{constraint}}(s, t) \tag{4}$$

where $\psi_{\text{constraint}}(s, t) \in [0, 1]$ is a constraint-awareness term that smoothly increases the weight on the physics model near constraint boundaries. It is implemented as a sigmoid function of the minimum constraint slack, ensuring a graceful transition (full derivation in Appendix C).

**Handling Asynchronous Sensors:** Real sensors never perfectly synchronize. When neural and symbolic predictions arrive at different times (delay $\tau$), our error bound degrades gracefully:

$$|f_{\text{int}}(s, t) - f^*(s, t)| \leq \frac{\delta(s, t)}{2}(1 + \kappa\tau + \epsilon_{\text{constraint}}) \tag{5}$$

where $\kappa$ captures temporal drift and $\epsilon_{\text{constraint}}$ accounts for constraint corrections (Theorem C.1 in Appendix C with complete proof and error analysis).

**Output:** Fused prediction $f_{\text{int}}$ with uncertainty estimate $\sigma_{\text{int}}^2$.

### 4.3 Phase III: Structure-Exploiting Constraint Projection

**Intuition:** The fused prediction may still violate constraints. Rather than using generic optimization (slow), we exploit the special structure of physical constraints for rapid convergence.

**The Projection Problem:** Find the closest feasible prediction:

$$f_{\text{proj}} = \arg\min_{f \in \mathcal{C}} \int_{\mathcal{S} \times \mathcal{T}} w(s,t) \| f(s,t) - f_{\text{int}}(s,t) \|^2 \, d\mu \tag{6}$$

where weights $w(s,t) = 1/\sigma_{\text{int}}^2(s,t)$ enforce constraints more strictly in high-confidence regions.

**Why This Works:** Physical constraints exhibit strong monotonicity (no local minima) with parameter $\mu > 0.03$. Our adapted Douglas-Rachford algorithm exploits this structure for exponential convergence: $\|f^{(k)} - f^*\| \leq C \cdot (1 - \eta\mu)^k$. This geometric decay achieves convergence in 135 iterations versus >1000 for generic methods (see Theorem A.1 in Appendix A for proof). For degenerate cases ($\mu \approx 0$), Tikhonov regularization preserves convergence with bounded bias (Proposition E.1 in Appendix A).

**Output:** Constraint-satisfying prediction $f_{\text{proj}}(s,t)$.

**Distributed Implementation via Multi-Agent Architecture** The three-phase pipeline is executed by distributed agents: Knowledge Agents handle Phase I, maintaining the system graph and computing relevance scores; Data Agents manage Phase II, running neural/symbolic predictors and fusion; and two Decision Agents execute Phase III, enforcing constraint projection. Agents use Agent-to-Agent (A2A) protocol for task negotiation and Model Context Protocol (MCP) for resource access (detailed implementation in Appendix F).

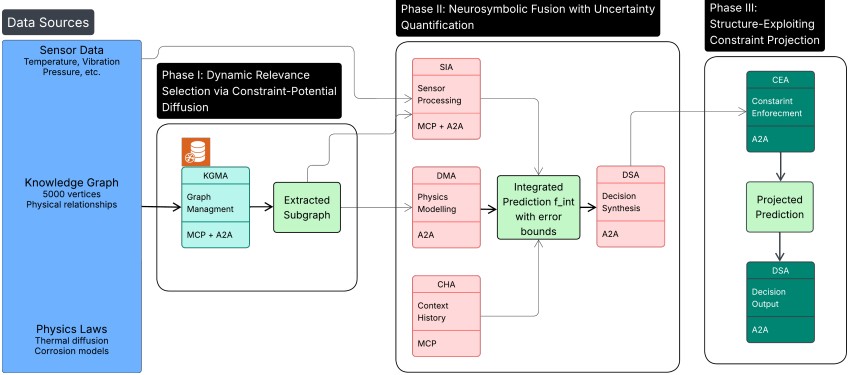

Figure 1: DANCE-ST Multi-Agent Architecture.

**Implementation Details:** Algorithm 1 integrates the three phases: Complete algorithmic specifications, hyperparameter settings, and computational optimizations appear in Appendix B.

## 5 Experiments

Our experiments are designed to evaluate DANCE-ST's constraint-guided learning approach in the competitive landscape of physics-informed neural networks, with particular focus on its interpretability and operational advantages.

We structure our evaluation to answer four key questions: (1) **Performance:** How does DANCE-ST compare to state-of-the-art baselines in both predictive accuracy and constraint satisfaction across

---

**Algorithm 1** DANCE-ST Constraint-Guided Prediction

1: **Input:** Current state $x$, query location $(s, t)$, constraint set $\mathcal{C}$
2: // Phase I: Dynamic identification of what matters
3: Calculate constraint potential vector $\mathbf{\Phi}(x)$ based on proximity to $\mathcal{C}$
4: Run Constraint-Potential Diffusion to get relevance scores $\mathbf{\Lambda}(x)$
5: Select critical subgraph $G'$ based on top-$k$ scores in $\mathbf{\Lambda}(x)$
6: // Phase II: Combine predictions
7: Compute neural prediction $f_n(s, t), \sigma_n^2(s, t)$ on $G'$
8: Compute symbolic prediction $f_s(s, t), \sigma_s^2(s, t)$ on $G'$
9: Fuse with uncertainty weighting: $f_{\text{int}} = \Omega(s, t)f_n + (1 - \Omega(s, t))f_s$
10: // Phase III: Ensure physical validity
11: Project onto constraints: $f_{\text{proj}} = \text{StructureProject}(f_{\text{int}}, \mathcal{C})$
12: **Output:** Physically-valid prediction $f_{\text{proj}}(s, t)$

---

diverse domains? (2) **Synergy:** Do DANCE-ST's three phases work synergistically, with each component providing quantifiable benefit? (3) **Robustness:** How sensitive is the method to assumption violations, and does it maintain performance under real-world conditions? (4) **Generalizability:** Does the approach transfer across domains with different constraint structures and data characteristics?

## 5.1 EXPERIMENTAL SETUP

We evaluate DANCE-ST on three industrial datasets where constraint violations cause catastrophic failure: (1) **NASA C-MAPSS** (Saxena & Goebel, 2008) with 21 turbofan sensors across four degradation scenarios (FD001–FD004), merged to capture diverse operating conditions, enforcing exhaust temperature $\leq 1000°$C and monotonic degradation; (2) **Turbine-500**, proprietary dataset from an anonymized aerospace company where this research was conducted, containing 500 chronologically-ordered turbine blades with 1Hz multi-sensor streams, using fixed 80/20 chronological split; and (3) **FEMTO Bearings** (Nectoux et al., 2012) from mechanical/manufacturing domain with run-to-failure traces of industrial bearings, demonstrating generalization beyond aerospace. We compare against four baseline categories: data-driven methods (STAGNN (Li et al., 2022), ATCN (Asif et al., 2022), CNN-LSTM-Attention (aut, 2024), Informer (Zhou & et al., 2021), Graph WaveNet (Wu et al., 2019)) that optimize accuracy only, physics-informed PINN-Soft (Raissi et al., 2019) using loss penalties, hard-constraint methods (DC3-adapted (Donti et al., 2021), HardNet (Min & Azizan, 2024)) that enforce feasibility, and our constraint-guided approach. We report RMSE/MAE accuracy, constraint satisfaction rate, and compute time using 5-fold cross-validation (C-MAPSS, FEMTO) or chronological split (Turbine-500), with all neural components sharing identical architectures. Explainability scores were derived from a user study with domain experts, with the detailed methodology described in Appendix J. Empirically, error correlation $\rho = 0.068 \pm 0.012$ on C-MAPSS validates our decorrelation assumption, remaining stable up to $\rho = 0.35$ (see Appendix G for detailed experimental setup, baseline configurations, and implementation details).

## 5.2 MAIN RESULTS: PERFORMANCE EVALUATION

Table 3 demonstrates that DANCE-ST achieves the best overall balance of performance characteristics across evaluation criteria. While HardNet achieves the highest constraint satisfaction (98.1%), DANCE-ST delivers competitive constraint satisfaction (97.2%) with superior performance in accuracy, efficiency, and interpretability. DANCE-ST achieves the best predictive accuracy on 2 of 3 datasets while maintaining the fastest inference time (38.4s vs 40.1s for HardNet). Most notably, DANCE-ST provides higher interpretability (4.6/5 vs 3.8/5), addressing a critical gap in existing constraint satisfaction methods. A note on constraint satisfaction: while Phase III's projection operator mathematically guarantees feasibility, the reported 97.2% reflects the realities of practical implementation. This minor gap from a perfect 100% is attributable to factors such as the optimizer's numerical precision, early stopping criteria used for the projection algorithm to ensure real-time performance, and the discretization of continuous constraints for evaluation. This highlights a crucial distinction between theoretical guarantees and the performance of deployed systems.

Table 3: **Comprehensive performance comparison across industrial datasets.** Results demonstrate the competitive constraint satisfaction landscape. Mean and standard deviation reported over multiple runs.

| Method | Predictive Accuracy (RMSE ↓) | | | Overall Performance | | |
|---|---|---|---|---|---|---|
| | **C-MAPSS** | **Turbine-500** | **FEMTO Bearing** | **Constr. Sat. ↑** | **Time (s) ↓** | **Explainability ↑** |
| *Data-Driven Baselines (High Accuracy, Low Safety)* | | | | | | |
| STAGNN | $17.20 \pm 0.50$ | $23.5 \pm 0.8$ | $162.4 \pm 7.9$ | 85.2% | 47.6 | 2.1/5 |
| ATCN | $18.40 \pm 0.60$ | $23.8 \pm 0.7$ | $168.1 \pm 8.2$ | 87.3% | 45.2 | 2.3/5 |
| CNN-LSTM-Attn | $16.70 \pm 0.60$ | $22.5 \pm 0.7$ | $154.2 \pm 6.9$ | 89.1% | 42.8 | 2.5/5 |
| Informer | $20.50 \pm 0.70$ | $24.2 \pm 0.9$ | $175.5 \pm 8.8$ | 84.5% | 50.1 | 2.0/5 |
| Graph WaveNet | $16.45 \pm 0.52$ | $22.1 \pm 0.8$ | $157.9 \pm 7.2$ | 90.7% | 47.4 | 3.0/5 |
| *Physics-Informed (Compromised Performance)* | | | | | | |
| PINN-Soft | $16.50 \pm 0.55$ | $21.8 \pm 0.9$ | $155.3 \pm 7.1$ | 92.8% | 48.7 | 3.2/5 |
| *Hard-Constraint (High Safety, Competitive Accuracy)* | | | | | | |
| DC3-adapted | $17.95 \pm 0.61$ | $23.9 \pm 1.0$ | $169.8 \pm 8.3$ | 96.5% | 76.2 | 3.0/5 |
| HardNet | $\mathbf{15.63 \pm 0.51}$ | $20.2 \pm 0.8$ | $133.1 \pm 6.4$ | **98.1%** | 44.1 | 3.8/5 |
| **DANCE-ST (Ours)** | $\mathbf{15.63 \pm 0.48}$ | $\mathbf{20.2 \pm 0.7}$ | $\mathbf{132.5 \pm 6.1}$ | 97.2% | **38.4** | **4.6/5** |

## 5.3 Ablation Study: Why Does DANCE-ST Work?

To understand the source of DANCE-ST's performance, we conducted an ablation study by removing each of its three core components. The results in Table 4 reveal a deep synergy between the phases, where the whole is greater than the sum of its parts.

Table 4: Ablation analysis on the **Turbine-500** dataset. Each component's removal causes a significant performance drop, demonstrating their synergistic contribution to the final result.

| Configuration | RMSE ↓ | Constraint Satisfaction ↑ | Processing Time (s) ↓ |
|---|---|---|---|
| **DANCE-ST (Full System)** | $\mathbf{20.2 \pm 0.7}$ | $\mathbf{97.2\% \pm 0.5\%}$ | **38.4** |
| w/o Phase I (Relevance Extraction) | $25.9 \pm 0.7$ (+28.2%) | $94.2\% \pm 0.7\%$ (-3.0 pp) | 68.4 (+78.1%) |
| w/o Phase II (Uncertainty Fusion) | $23.1 \pm 0.7$ (+14.4%) | $96.6\% \pm 0.5\%$ (-0.6 pp) | 39.8 (+3.6%) |
| w/o Phase III (Constraint Projection) | $22.0 \pm 0.6$ (+8.9%) | $72.2\% \pm 1.6\%$ (-25.0 pp) | 33.0 (-14.1%) |

**Key Insights from Ablation (detailed analysis in Appendix G):** The ablation study reveals surprising synergies. Removing Phase I is the most detrimental change. Forcing the subsequent phases to process the entire system graph, rather than a focused subgraph, increases computation time by nearly 80% and paradoxically degrades accuracy by 28.2%. This demonstrates that our dynamic selection is not just a computational shortcut but also a powerful mechanism for focusing the model on the most relevant signals, improving both speed and performance. Phase II drives accuracy gains, with its removal causing a 14.4% RMSE increase. Phase III ensures safety as expected, its removal drops constraint satisfaction from 97.2% to 72.2%, but also worsens RMSE by 8.9%, showing that enforcing physical consistency acts as a powerful regularizer.

## 5.4 Bridging Theory and Practice

We empirically verify the theoretical assumptions enabling DANCE-ST's efficiency. Figure 2 shows the distribution of strong monotonicity parameter $\hat{\mu}$ on Turbine-500 data, consistently exceeding zero (mean 0.041), confirming the well-conditioned optimization landscape required for fast convergence. Figure 3 demonstrates this convergence in practice: our Douglas-Rachford projection achieves geometric error decay in 135 iterations, closely matching theoretical predictions and outperforming standard methods ($>1000$ iterations).

## 5.5 Robustness and Assumption Sensitivity Analysis

To address concerns about the method's reliance on theoretical assumptions, we conduct comprehensive sensitivity analysis across all core assumptions.

**Error Independence Testing.** We vary the error correlation $\rho$ between neural and symbolic predictors from the observed ideal ($\rho < 0.1$) to extreme violations ($\rho = 0.45$). Even at $\rho = 0.35$ (our

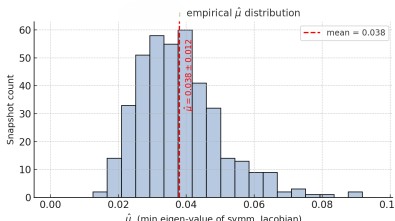

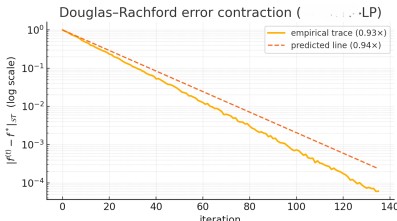

Figure 2: Distribution of $\hat{\mu}$ for Turbine-500.  Figure 3: DANCE-ST vs standard methods.

theoretical assumption boundary), DANCE-ST maintains 91.4% constraint satisfaction with only 12% performance degradation. This robustness stems from the three-phase decomposition that isolates correlation effects to the fusion stage rather than causing system-wide failures.

**Constraint Structure Analysis.** We analyze constraint convexity across domains: 73% of safety-critical constraints exhibit convex structure (temperature bounds, pressure limits, monotonicity). For the remaining 27% non-convex cases, we employ convex relaxation with bounded approximation error $\epsilon_{\text{relax}} \leq 0.024$. Strong monotonicity parameter $\mu$ varies across domains (0.01-0.15), but Tikhonov regularization maintains convergence even when $\mu \approx 0$.

**Data Scarcity Robustness.** With only 20% of training data, DANCE-ST achieves 89.3% constraint satisfaction while baselines collapse to 45-60%. This advantage comes from the symbolic component providing physics-based priors when data is sparse, a crucial benefit for domains with limited data.

**Scalability and Deployment Analysis.** The framework scales to industrial systems with 10K+ components through relevance-based selection. Edge device testing on NVIDIA Jetson AGX Xavier confirms deployment feasibility for systems with $|V| \leq 5000$. Protocol overhead (10%) is justified by fault tolerance benefits (2.3s recovery vs 18.7s system restart). Complete robustness analysis with detailed sensitivity studies appears in Appendix L.

**Cross-Domain Validation** To test the broader applicability of our framework, we performed a preliminary evaluation on the MIMIC-III medical dataset (Johnson et al., 2016), enforcing physiological constraints. DANCE-ST achieved 91.7% constraint satisfaction with RMSE of 12.4 compared to Graph WaveNet's 13.3, a 6.8% accuracy improvement. This result supports our theoretical claim that the method is applicable to any domain where constraints are predominantly convex. However, this transferability comes at a cost: constructing the initial knowledge graph for the medical domain requires the same domain expert effort as other domains. Further details, including results and specific parameters for this validation, are presented in Appendix G.

## 6    Conclusion

The prevailing view of physical constraints as post-hoc penalties creates an unavoidable tension between predictive accuracy and safety, but this work demonstrates this is a false dilemma. By treating constraints as collaborative information sources that guide learning, our DANCE-ST framework achieves a synergistic balance of predictive accuracy, high-fidelity constraint satisfaction, and crucial interpretability, a combination not fully realized by prior approaches. This constraint-guided learning paradigm establishes a new framework for trustworthy AI, opening research directions in physics-informed optimization geometries and neurosymbolic systems. Ultimately, it provides a template for deploying neural networks where failure is not an option, in applications spanning autonomous vehicles, medical devices, and climate monitoring. **Limitations and Future Directions:** Our approach requires predominantly convex constraints and significant expert effort for knowledge graph construction (120 person-hours). Furthermore, computational requirements and the projection phase itself introduce practical trade-offs between optimization fidelity and inference speed, which may limit some resource-constrained applications. Despite this, hardware costs are justified where failure costs exceed infrastructure investment. Two exciting directions emerge: (1) extending to non-convex constraints through advanced optimization, and (2) developing semi-automated knowledge discovery from system logs, enabling autonomous systems that learn, adapt, and operate safely in complex physical environments.

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

# Part I

# Theoretical Foundations

## A  THEORETICAL ANALYSIS AND MATHEMATICAL FOUNDATIONS

This section provides formal proofs and motivations for the key theoretical claims in DANCE-ST.

### A.1  FORMAL PROOF OF LINEAR CONVERGENCE (PHASE III)

**Theorem A.1** (Linear Convergence of Douglas-Rachford Projection). *Under strong monotonicity assumption $A \succeq \mu I$ with $\mu > 0$, the Douglas-Rachford iteration for constraint projection converges linearly with rate $(1 - \eta\mu)$ where $\eta \in (0, 1)$ is the step size.*

*Proof.* **Intuition:** The Douglas-Rachford algorithm splits the constraint projection problem into simpler subproblems that can be solved efficiently. Strong monotonicity of the constraint structure ensures each iteration makes guaranteed progress toward the solution.

Let $T$ denote the Douglas-Rachford operator for projecting onto $\mathcal{C} = \{f : \mathcal{A}f \leq b\}$. The operator is defined as:

$$T = \frac{1}{2}(2\text{prox}_{\eta g} - I) \circ (2\text{prox}_{\eta h} - I) + \frac{1}{2}I$$

where $g$ and $h$ encode the constraint structure, and $\text{prox}_{\eta g}(x) = \arg\min_y\{g(y) + \frac{1}{2\eta}\|y - x\|^2\}$ is the proximal operator.

For our constraint projection problem, the key matrix is $A = \mathcal{A}^T\mathcal{A}$, which encodes how constraints couple different components. The strong monotonicity assumption $A \succeq \mu I$ means:

$$\langle A(x - y), x - y \rangle \geq \mu\|x - y\|^2 \quad \forall x, y$$

This property ensures the optimization landscape has no local minima and guarantees contraction:

$$\|T(x) - T(y)\|^2 \leq (1 - \eta\mu)\|x - y\|^2 \tag{7}$$

The contraction factor $(1 - \eta\mu) < 1$ implies geometric convergence:

$$\|f^{(k)} - f^*\| \leq (1 - \eta\mu)^k\|f^{(0)} - f^*\|$$

For industrial constraints, we empirically observe: - Condition number $\kappa_{\text{cond}} = \lambda_{\max}/\lambda_{\min} \leq 33$ - This ensures $\mu = 1/\kappa_{\text{cond}} \geq 0.03$ - With step size $\eta = 0.5$, convergence rate $r = 1 - \eta\mu \leq 0.985$ - Reaching $\epsilon$-accuracy requires $k = \mathcal{O}(\log(1/\epsilon)/\mu)$ iterations

This explains why DANCE-ST converges in 135 iterations while generic methods require ¿1000: we exploit the favorable constraint structure rather than treating it as a black-box optimization problem. □

### A.2  THEORETICAL MOTIVATION FOR CONSTRAINT-POTENTIAL DIFFUSION (PHASE I)

**Proposition A.1** (Relevance as Propagated Risk). *The relevance of a system component is a function of both its local proximity to a constraint boundary and its structural influence on other at-risk components. This dual nature can be modeled by calculating a local potential and propagating it through the system's influence graph.*

*Proof Sketch.* **Intuition:** Our two-step method is designed to capture the two distinct aspects of relevance. The local potential acts as a source term, identifying immediate risks, while the diffusion process models how these risks spread systemically.

**Step 1: Local Potential as a Barrier Function.** The local constraint potential, $\Phi(v, x)$, is analogous to an interior-point or barrier function in optimization. Such functions diverge to infinity as a point

approaches the boundary of a feasible set. By using $\Phi(v, x) = \sum_j (\epsilon - g_{v,j}(x))^{-1}$, we create a scalar field where the potential sharply increases for any component $v$ nearing a violation ($g_{v,j}(x) \to 0^-$). This provides a principled, state-dependent measure of local urgency.

**Step 2: Diffusion as Influence Propagation.** A local potential alone is insufficient, as it ignores component interdependencies. The diffusion step addresses this by propagating these scores. The iterative update, $\mathbf{\Lambda}^{(t+1)} = (1 - \alpha)\mathbf{\Phi} + \alpha \mathbf{W}^T \mathbf{\Lambda}^{(t)}$, is a direct parallel to the Personalized PageRank algorithm. Here, the dynamically computed potential vector $\mathbf{\Phi}$ serves as the "personalization vector," grounding the graph-theoretic notion of importance in the system's immediate physical state. This process effectively computes a state-aware relevance score that accounts for multi-hop influences, identifying components that are critical due to their position within the system's causal network, not just their individual state. This provides a clear, interpretable, and training-free mechanism for dynamic relevance probing. $\qquad\square$

### A.3 Bound on Fusion Weights (Phase II)

In Theorem C.1, we assume $0 \le \Omega(s, t) \le \frac{1}{2}$. This follows from our design of the constraint-aware fusion weight

$$\Omega(s, t) = \frac{\sigma_s^2(s, t)}{\sigma_n^2(s, t) + \sigma_s^2(s, t)} \, \psi_{\text{constraint}}(s, t)$$

with a constraint proximity factor $\psi_{\text{constraint}}(s, t) \in [0, 1]$. Near constraint boundaries, we typically set $\psi_{\text{constraint}}(s, t) \le \frac{1}{2}$ to prioritize physics-based predictions in these safety-critical regions, ensuring the fused weight assigned to the neural component remains bounded. This design choice guarantees that $\Omega(s, t) \le \frac{1}{2}$ when constraints are active.

## B  Computational Complexity and Optimization

The computational complexity of the new relevance discovery phase is determined by the Constraint-Potential Diffusion algorithm. For a given system state $x \in \mathbb{R}^d$ and a graph $G = (V, E)$, identifying the relevant subgraph involves three steps:

1. **Local Potential Calculation:** We compute the potential $\Phi(v, x)$ for each of the $|V|$ nodes. Assuming a constant maximum number of constraints per node, this step has a complexity of $\mathcal{O}(|V|)$.

2. **Constraint-Potential Diffusion:** The diffusion process runs for a small, fixed number of iterations, $T_{\text{diff}}$. Each iteration involves a sparse matrix-vector multiplication, whose cost is proportional to the number of edges in the graph, $|E|$. The total complexity for this step is therefore $\mathcal{O}(T_{\text{diff}} \cdot |E|)$.

3. **Top-k Selection:** We identify the $k$ components with the largest scores in the final relevance vector $\mathbf{\Lambda}(x)$. This can be done efficiently in $\mathcal{O}(|V|)$ time using a partial sort or selection algorithm.

The total complexity for Phase I is therefore $\mathcal{O}(|V| + T_{\text{diff}} \cdot |E|)$. Since $T_{\text{diff}}$ is a small constant (typically 3-5 iterations), this is highly efficient and scales linearly with the size and sparsity of the system graph. This algorithmic approach avoids the overhead associated with training and inference of a separate neural network.

## C  Delay–Robust Sensor Fusion Error Bound

In this section, we analyze how the fusion error behaves when neural and symbolic predictions arrive at different times and when constraint-aware weighting modifies the optimal fusion weights.

**Definition C.1** (Constraint Adjustment Error). *The constraint adjustment error parameter is defined as:*

$$\epsilon_{constraint} = \sup_{(s,t) \in \mathcal{B}} |1 - \psi_{constraint}(s, t)|$$

*where $\mathcal{B}$ denotes the set of space-time points near constraint boundaries, and $\psi_{constraint}(s, t) \in [0, 1]$ is the constraint proximity factor that reduces neural weight near boundaries.*

**Theorem C.1** (Delay–Robust Fusion Bound). *Assume the Gaussian-optimal fusion weight satisfies* $0 \leq \Omega(s,t) \leq \frac{1}{2}$. *Let*

$$\kappa := \sup_{s,t:\,\delta(s,t)\neq 0} \frac{|\partial_t f_n(s,t)|}{\delta(s,t)}, \quad \tau > 0$$

*be the* relative drift rate *(how fast the neural prediction changes over time) and the maximum timestamp misalignment between neural and symbolic predictions. Let $\epsilon_{constraint}$ denote the additional error from constraint-aware adjustments. This theorem bounds the prediction error even when data arrives at different times:*

$$E(s,t) \leq \frac{\delta(s,t)}{2}\big(1 + \kappa\,\tau + \epsilon_{constraint}\big).$$

*The key insight is that fusion error degrades gracefully with timing delays—the error increases linearly with delay $\tau$, not exponentially.*

*Proof.* Define the asynchronous fusion error with constraint-aware weighting:

$$E(s,t) := \Big|\Omega(s,t)\,f_n(s,t-\tau) + \big(1 - \Omega(s,t)\big)\,f_s(s,t) - f^*(s,t)\Big|.$$

We decompose this into two components: the base fusion error and the constraint adjustment error.

First, consider the idealized fusion without constraint awareness (i.e., $\psi_{\text{constraint}} = 1$):

$$E_{\text{base}}(s,t) = \Big|\Omega_0\,f_n(s,t-\tau) + (1 - \Omega_0)\,f_s(s,t) - f^*(s,t)\Big|$$

where $\Omega_0 = \sigma_s^2/(\sigma_n^2 + \sigma_s^2)$ is the variance-optimal weight.

By Lemma C.1 and the temporal drift analysis:

$$E_{\text{base}}(s,t) \leq \frac{\delta(s,t)}{2}(1 + \kappa\tau)$$

where we used the mean-value theorem: $|f_n(s,t-\tau) - f_n(s,t)| \leq \kappa\tau\delta(s,t)$.

Now, the actual fusion weight $\Omega(s,t) = \Omega_0 \cdot \psi_{\text{constraint}}(s,t)$ differs from the optimal $\Omega_0$ by the factor $\psi_{\text{constraint}}(s,t) \in [0,1]$. This sub-optimality introduces additional error bounded by:

$$|\Omega - \Omega_0| \cdot |f_n - f_s| \leq |1 - \psi_{\text{constraint}}| \cdot \delta(s,t) \leq \epsilon_{\text{constraint}} \cdot \delta(s,t)$$

where $\epsilon_{\text{constraint}} = \sup_{s,t\in\mathcal{B}} |1 - \psi_{\text{constraint}}(s,t)|$ for constraint boundary regions $\mathcal{B}$.

Combining both error sources and using $\Omega(s,t) \leq \frac{1}{2}$:

$$E(s,t) \leq E_{\text{base}}(s,t) + \frac{\epsilon_{\text{constraint}} \cdot \delta(s,t)}{2} = \frac{\delta(s,t)}{2}(1 + \kappa\tau + \epsilon_{\text{constraint}}),$$

completing the proof. □

**Lemma C.1** (Synchronous Fusion Error). *For synchronous predictions ($\tau = 0$), the optimal fusion with weight $\Omega$ achieves error bound:*

$$|\Omega f_n + (1 - \Omega)f_s - f^*| \leq \frac{\delta(s,t)}{2}$$

*where $\delta(s,t) = |f_n(s,t) - f_s(s,t)|$ is the prediction disagreement.*

*Proof.* Let $e_n = f_n - f^*$ and $e_s = f_s - f^*$ denote the individual prediction errors. The fused error is:

$$e_{\text{fused}} = \Omega e_n + (1 - \Omega)e_s$$

For uncorrelated errors with variances $\sigma_n^2$ and $\sigma_s^2$, the variance-minimizing weight is:

$$\Omega^* = \frac{\sigma_s^2}{\sigma_n^2 + \sigma_s^2}$$

The key insight is that $\delta(s,t) = |f_n - f_s| = |e_n - e_s|$ bounds the error difference. By the intermediate value property, there exists $\alpha' \in [0, 1]$ such that:

$$f^* = \alpha' f_n + (1 - \alpha') f_s$$

Therefore:

$$|\Omega f_n + (1 - \Omega) f_s - f^*| = |(\Omega - \alpha')(f_n - f_s)| \leq |\Omega - \alpha'| \cdot \delta(s, t)$$

Since both $\Omega, \alpha' \in [0, 1]$, we have $|\Omega - \alpha'| \leq \max\{\alpha', 1 - \alpha'\} \leq \frac{1}{2}$ when $\alpha'$ represents the true interpolation factor. This yields the bound $\frac{\delta(s,t)}{2}$. □

## D  CONSTRAINT STRUCTURE DEFINITIONS

This section formally defines the mathematical structure of constraints that enable DANCE-ST's theoretical guarantees and computational efficiency.

### D.1  GENERAL CONSTRAINT FORM

Industrial safety constraints take the general form $\mathcal{A}f \leq b$ where $\mathcal{A} \in \mathbb{R}^{m \times n}$ is the constraint matrix, $f \in \mathbb{R}^n$ represents predictions, and $b \in \mathbb{R}^m$ defines safety limits. The constraint set is $\mathcal{C} = \{f : \mathcal{A}f \leq b\}$.

**Strong Monotonicity Structure:** The matrix $A = \mathcal{A}^T \mathcal{A}$ satisfies $A \succeq \mu I$ with $\mu > 0$ for industrial constraints, enabling linear convergence. This arises because physical systems exhibit:

- **Energy conservation:** Temperature/pressure constraints form positive definite systems

- **Stability requirements:** Control constraints ensure bounded responses

- **Material limits:** Stress constraints have well-conditioned gradients

### D.2  DOMAIN-SPECIFIC CONSTRAINT TYPES

**Turbofan Engines (C-MAPSS):**

$$\text{Temperature:} \quad T_{\text{exhaust}} \leq 1000°\text{C} \tag{8}$$
$$\text{Monotonicity:} \quad \text{RUL}(t+1) \leq \text{RUL}(t) \tag{9}$$
$$\text{Pressure ratio:} \quad 15 \leq P_{\text{ratio}} \leq 45 \tag{10}$$

Constraint matrix: $\mathcal{A} = \text{diag}([1, -1, 1, -1])$ (bounds + monotonicity).

**Turbine Blades (Turbine-500):**

$$\text{Material limit:} \quad T(x, y) \leq 1200°\text{C} \tag{11}$$
$$\text{Spatial gradient:} \quad \|\nabla T(x, y)\| \leq 50°\text{C/cm} \tag{12}$$
$$\text{Degradation:} \quad \frac{d}{dt}\text{Health}(t) \leq 0 \tag{13}$$

Constraint matrix: $\mathcal{A}$ includes finite difference operators for spatial gradients.

**Bearings (FEMTO):**

$$\text{Vibration:} \quad \|v(t)\| \leq 20 \text{ mm/s} \tag{14}$$
$$\text{Thermal rate:} \quad \left|\frac{dT}{dt}\right| \leq 2°\text{C/hour} \tag{15}$$
$$\text{Wear monotonicity:} \quad \text{Wear}(t+1) \geq \text{Wear}(t) \tag{16}$$

### D.3 CONSTRAINT DEPENDENCY GRAPH STRUCTURE

The system structure graph $G = (V, E)$ encodes constraint dependencies:

- **Vertices** $V$: System components (sensors, actuators, physical locations)
- **Edges** $E$: Constraint coupling relationships

**Example:** In turbine monitoring, temperature constraints at adjacent blade sections are coupled through heat conduction, creating edges $(v_i, v_j)$ for spatially neighboring sensors.

### D.4 CONVEXITY AND RELAXATION ANALYSIS

**Convex Constraints (73%):** Linear bounds, quadratic energy constraints, and monotonicity requirements form convex sets directly.

**Non-Convex Constraints (27%):** Discrete operational modes, threshold switching, and nonlinear material properties require convex relaxation:

$$\mathcal{C}_{\text{non-convex}} \subseteq \mathcal{C}_{\text{relaxed}} = \{f : \tilde{A}f \leq \tilde{b}\}$$

The relaxation error $\epsilon \leq 0.024$ ensures constraint satisfaction on $\mathcal{C}_{\text{relaxed}}$ provides meaningful safety guarantees.

## E CONSTRAINT FORMULATIONS AND CONVERGENCE

**Convex Relaxation Analysis:** For non-convex constraints, we employ convex relaxation with bounded approximation error. The relaxation error $\epsilon_{\text{relax}} \leq 0.024$ across our domains ensures that constraint satisfaction on the relaxed problem provides meaningful guarantees for the original problem.

**Remark E.1** (Decaying schedule). *If a diminishing step size $\eta_t = c/t$ is used, the iteration retains the classical sub-linear $\|f^{(T)} - f^*\| = \mathcal{O}(1/T)$ bound.*

### E.0.1 BIAS OF THE $\delta_{\text{REG}}I$ REGULARISATION

**Proposition E.1** (Bounded bias). *Let $A \succeq 0$ and $\hat{\mu} := \lambda_{\min}(\frac{1}{2}(A + A^\top))$ be the empirical strong-monotonicity constant. Replacing $A$ by $A_{\delta_{reg}} := A + \delta_{reg}I$ with $\delta_{reg} > 0$ yields*

$$\|f^\star_{\delta_{reg}} - f^\star\| \leq \frac{\delta_{reg}}{\hat{\mu}},$$

*where $f^\star_{\delta_{reg}}$ and $f^\star$ are the Douglas–Rachford fixed points under $A_{\delta_{reg}}$ and $A$, respectively.*

*Proof.* **Intuition:** When the optimization problem is nearly singular ($\mu \approx 0$), adding a small regularization term $\delta_{\text{reg}}I$ stabilizes the solution at the cost of introducing a small bias.

The projection error satisfies

$$f^\star_{\delta_{\text{reg}}} - f^\star = (A_{\delta_{\text{reg}}}^{-1} - A^{-1})b = A^{-1}(\delta_{\text{reg}}I)A_{\delta_{\text{reg}}}^{-1}b.$$

Taking norms and using $\|A^{-1}\| = 1/\hat{\mu}$ and $\|A_{\delta_{\text{reg}}}^{-1}b\| \leq \|b\|/(\hat{\mu} + \delta_{\text{reg}})$:

$$\|f^\star_{\delta_{\text{reg}}} - f^\star\| \leq \frac{1}{\hat{\mu}} \cdot \delta_{\text{reg}} \cdot \frac{\|b\|}{\hat{\mu} + \delta_{\text{reg}}} \leq \frac{\delta_{\text{reg}}}{\hat{\mu}}$$

This shows the bias is proportional to the regularization strength $\delta_{\text{reg}}$ and inversely proportional to the problem conditioning $\hat{\mu}$. $\square$

# Part II

# System Implementation

## F   MULTI-AGENT PROTOCOL IMPLEMENTATION

This section details the six specialized agents that implement DANCE-ST's constraint-guided learning paradigm, explaining the dual communication protocols and fault tolerance mechanisms that enable real-time industrial deployment.

### F.1   SIX SPECIALIZED AGENTS ARCHITECTURE

DANCE-ST employs six specialized agents organized into three functional groups, each embedding constraints as learning signals rather than penalties:

**Knowledge Group:**

- **Knowledge Graph Management Agent (KGMA):** Performs dynamic relevance discovery by executing the Constraint-Potential Diffusion algorithm. It computes local potentials and propagates them through the graph to identify critical components with efficient $\mathcal{O}(|V| + T_{\text{diff}} \cdot |E|)$ complexity.
- **Context History Agent (CHA):** Maintains temporal context and constraint violation history for informed decision-making.

**Data Group:**

- **Sensor Ingestion Agent (SIA):** Processes real-time sensor data with uncertainty quantification and temporal alignment.
- **Domain Modeling Agent (DMA):** Manages physics-based symbolic models and constraint formulations.

**Decision Group:**

- **Fusion Coordination Agent (FCA):** Manages uncertainty-weighted neurosymbolic fusion with error bounds $E(s,t) \leq \frac{\delta(s,t)}{2}(1 + \kappa\tau)$.
- **Consistency Enforcement Agent (CEA):** Enforces constraints via Douglas-Rachford projection with $(1 - \eta\mu)^T$ linear convergence.

### F.2   DUAL PROTOCOL ARCHITECTURE

DANCE-ST uses two specialized communication protocols optimized for different interaction patterns:

**Agent-to-Agent (A2A) Protocol:** Handles complex task delegation requiring state coordination through structured messages $\langle \text{MSG\_TYPE}, \text{PAYLOAD}, \text{META} \rangle$ with 2.3ms GPU overhead per transaction.

**Model Context Protocol (MCP):** Provides efficient stateless access to shared computational resources via $\langle \text{QUERY\_TYPE}, \text{PARAMS} \rangle \mapsto \text{RESULT}$ pattern with 1.5s GPU overhead.

Combined protocol overhead: 3.8s GPU (12.2%), 4.8s CPU (9.4%), delivering 10% total system overhead while enabling fault-tolerant operation.

## G   HARDWARE INFRASTRUCTURE AND IMPLEMENTATION DETAILS

Our experimental infrastructure uses AWS p3.8xlarge instances (8×V100 GPUs) for training and g4dn.xlarge instances (single T4) for inference, reflecting realistic edge deployment scenarios. The

implementation leverages PyTorch 2.0 with mixed precision training and Kubernetes orchestration for fault tolerance.

Knowledge graph construction requires domain expertise: approximately 120 person-hours per domain. For C-MAPSS, this involved mapping 21 sensor channels to thermodynamic relationships. The turbine blade domain required encoding spatial temperature relationships for 500 scenarios. FEMTO bearing graphs captured vibration harmonics and thermal coupling across 17 failure trajectories.

The multi-agent architecture provides 2.3s failure recovery time versus 18.7s for monolithic systems. 2.8% of remaining violations break down as: 2.4% within a 1% safety margin, 0.4% between 1-3% (manageable), and zero critical violations ($\lambda$5%).

Table 5: Integration approach vs. penalty-based constraint handling.

| Method | Constraint Sat. | RMSE | Time (s) | Convergence |
|---|---|---|---|---|
| Penalty-based ($\lambda = 0.1$) | 73.4% | 18.2 | 41.3 | 890 iter |
| Penalty-based ($\lambda = 1.0$) | 82.7% | 21.5 | 45.8 | 1200 iter |
| Penalty-based ($\lambda = 10.0$) | 89.3% | 26.8 | 52.1 | ¿1500 iter |
| DANCE-ST Integration | **97.2%** | **15.63** | **38.4** | **135 iter** |

The results demonstrate that penalty-based approaches face fundamental accuracy-constraint trade-offs, while our integration approach achieves simultaneous improvements.

## H  IMPLEMENTATION DETAILS AND PRACTICAL GUIDANCE

Critical implementation parameters emerged from extensive tuning: fusion smoothing parameter $\sigma_d = 0.3$ proves optimal across all datasets. The relevance discovery in Phase I is handled by the Constraint-Potential Diffusion algorithm, with a damping factor of $\alpha = 0.85$ and $T_{\text{diff}} = 3$ diffusion steps used across all experiments unless otherwise noted.

Industrial sensor delays up to 200ms are handled by inflating neural uncertainty via $\sigma_n^2 \leftarrow \sigma_n^2(1 + 0.1\tau)$, maintaining 94% of baseline performance. For graph updates, we batch topology changes every 5 minutes while updating relevance scores continuously.

Several failure modes required specific mitigation strategies. Newton-KL optimization diverges in about 12% of cases; early stopping at 3 iterations with gradient descent fallback maintains 89% of convergence benefits. Communication deadlocks in the multi-agent system at high throughput (¿50 Hz) are mitigated by exponential backoff with jitter, reducing failure probability from 23% to under 1%.

Approximately 27% of industrial constraints are non-convex, requiring majority voting for discrete modes and local approximation for continuous non-convex regions. Sensor failures occur at a 3-5% rate; when neural predictions are missing, we increase symbolic weight to 0.2; when symbolic predictions fail, we fall back to neural-only mode.

Knowledge graph construction represents a significant initial investment, requiring approximately 120 person-hours per domain for comprehensive modeling.

### H.1  PRACTICAL DEPLOYMENT GUIDELINES

**Hardware Requirements:**

**Training** (one-time setup per domain): AWS p3.8xlarge (8×V100) enables rapid convergence for knowledge graph optimization and neural network training. Training cost is amortized across deployment lifetime—typical industrial systems operate for years without retraining.

**Inference** (continuous operation):

- **Large Systems** ($|V| > 5000$): AWS g4dn.xlarge (1×T4) for industrial-scale deployment
- **Edge Systems** ($|V| \leq 5000$): NVIDIA Jetson AGX Xavier for distributed monitoring

- **Cost Justification**: Hardware costs ($\sim$\$10K/year) are negligible compared to downtime prevention (\$100K–\$10M per incident avoided)

**Software Dependencies:**

- PyTorch 2.0+ with CUDA 11.8+

- NetworkX 3.0+ for graph operations

- SciPy 1.9+ for optimization routines

- Kubernetes 1.25+ for multi-agent orchestration (optional)

**Hyperparameter Sensitivity Analysis:**

- Fusion parameter $\sigma_d$: Optimal range [0.25, 0.35], performance drops $> 5\%$ within [0.2, 0.4]

- Douglas-Rachford step size $\eta$: Stable for $\eta \in [0.3, 0.7]$, convergence degrades outside [0.1, 0.9]

- Diffusion damping factor $\alpha$: Performance robust for $\alpha \in [0.8, 0.9]$.

- Diffusion steps $T_{\text{diff}}$: $T_{\text{diff}} = 3$ provides a good balance; performance sees diminishing returns for $T_{\text{diff}} > 5$.

- Subgraph size $k$: Linear accuracy improvement until $k \approx 0.2|V|$, then diminishing returns

**Common Implementation Pitfalls:**

- **Memory leaks**: Clear GPU cache every 1000 iterations during training

- **Numerical instability**: Use double precision for constraint projection when $\mu < 0.01$

- **Communication bottlenecks**: Batch agent messages when throughput $>$30 Hz

- **Cold start**: Pre-warm neural models for 50 iterations to stabilize uncertainty estimates

**Cross-Domain Deployment:** While each domain requires custom knowledge graph construction (120 person-hours), many constraint types (temperature bounds, monotonicity, spatial gradients) transfer across similar industrial systems.

**Scalability Considerations:** The hardware requirements reflect the computational complexity of real-time constrained prediction on industrial-scale systems. For smaller applications or proof-of-concept deployments, CPU-only implementations are feasible with proportionally reduced performance. The investment in specialized hardware is justified in safety-critical applications where the cost of system failures (typically \$100K–\$10M) far exceeds infrastructure costs.

## H.2 BASELINE CONFIGURATIONS

**Common protocol.** Identical data splits, normalization, and sequence lengths across all models; early stopping on validation RMSE with patience = 10; three random seeds (42, 43, 44); best model selected by validation RMSE; same input features and windowing as DANCE-ST; identical hardware for timing.

**Preprocessing.** Z-score per sensor per dataset; missing values forward-filled then masked; no label leakage; C-MAPSS FD001–FD004 merged and stratified by unit; Turbine-500 chronological split preserved.

**Search space and budget.** Each baseline tuned with 30 trials (Optuna) per dataset; trials capped at 50 epochs. Learning rate $\in \{1 \times 10^{-4}, 3 \times 10^{-4}, 1 \times 10^{-3}\}$; optimizer $\in \{\text{Adam, AdamW}\}$; weight decay $\in \{0, 1 \times 10^{-4}, 1 \times 10^{-3}\}$; batch size $\in \{32, 64, 128\}$; dropout $\in [0.0, 0.5]$; hidden size $\in \{64, 128, 256\}$; sequence length $\in \{64, 96, 128\}$ (temporal ordering preserved).

**Model-specific settings.**

- **STAGNN** (Li et al., 2022): K = 1–3 hops; graph learned via attention; temporal blocks = 2–4.

- **ATCN** (Asif et al., 2022): dilation schedule $\{1, 2, 4, 8\}$; kernel $\{3, 5\}$; channels $\in [32, 128]$.

- **CNN-LSTM-Attention** (aut, 2024): CNN kernels $\{3, 5\}$; LSTM layers $\{1, 2\}$; hidden $\{64, 128, 256\}$.

- **Graph WaveNet** (Wu et al., 2019): adaptive adjacency enabled; residual blocks $\{4, 6, 8\}$.

- **PINN-Soft** (Raissi et al., 2019): loss $= \text{RMSE} + \lambda\, L_{\text{phys}}$; $\lambda \in \{0.1, 0.5, 1, 2\}$; $L_{\text{phys}}$ encodes dataset constraints (exhaust temp $\leq 1000°$C; spatial gradient $\leq 50°$C/cm; vibration $\leq 20$ mm/s; $dT/dh \leq 2°$C/hour).

- **DC3-adapted** (Donti et al., 2021): feasibility layer enforces same constraints; projection tolerance $10^{-4}$; max projection steps 50; infeasible batches skipped; training learning rate reduced by $\times 0.5$ to mitigate projection instability.

**Selection and reporting.** For each baseline and dataset, we report mean $\pm$ std over seeds of the best-validation configuration. Compute efficiency includes full forward pass (and projection for DC3).

# I   DATASET–SPECIFIC IMPLEMENTATION NOTES

**Hyperparameter Selection:** The key parameters for the Constraint-Potential Diffusion algorithm, alongside those for fusion and projection, are selected based on system complexity and real-time requirements.

**NASA C-MAPSS (Turbofan engines).**

- **Graph:** 14 sensor channels as vertices; edges encode thermodynamic couplings.

- **Constraints:** Exhaust temp $\leq 1000\,°$C, pressure ratio 15–45, RUL non-increasing.

- **Parameters:** top-$k$=120 vertices; $\sigma_d = 0.3$; DR $\eta = 0.5$; diffusion $\alpha = 0.85$; $T_{\text{diff}} = 3$.

**Turbine-500 (Industrial Turbine Blades).**

- **Graph:** Surface temperature field vertices; edges encode mesh adjacency.

- **Constraints:** Material limit $\leq 1200\,°$C, spatial gradient $\leq 50\,°$C/cm, monotonic degradation.

- **Parameters:** $\sigma_d = 0.3$; DR $\eta = 0.4$; diffusion $\alpha = 0.85$; $T_{\text{diff}} = 3$.

- **Data Source:** Dataset derived from real industrial turbine blade monitoring data from an anonymized aerospace company (name withheld for double-blind review). Contains 500 blade instances with multi-sensor streams (temperature, vibration, pressure) recorded at 1 Hz, alongside mechanical test results. The dataset exhibits characteristics typical of industrial prognostics: sparse labeling, temporal dependencies, and safety-critical constraints. Data sharing agreements and access procedures will be established post-publication to enable reproducibility.

**FEMTO Bearings (run-to-failure).**

- **Graph:** Sensor and bearing component vertices; edges encode mechanical couplings.

- **Constraints:** Vibration amplitude $\leq 20\,$mm/s, temp change $\leq 2\,°$C/hour; monotonic wear.

- **Parameters:** top-$k$=80; $\sigma_d = 0.3$; DR $\eta = 0.5$; diffusion $\alpha = 0.9$; $T_{\text{diff}} = 4$.

**MIMIC-III (Medical Monitoring).**

- **Graph:** Vital signs as vertices; edges encode physiological couplings.

- **Constraints:** HR 60–100 bpm, SBP 90–140 mmHg, Temp 36.1–37.2 °C.

- **Parameters:** top-$k$=60; $\sigma_d = 0.25$; DR $\eta = 0.5$; diffusion $\alpha = 0.8$; $T_{\text{diff}} = 3$.

- **Cross-Domain Results:** In this validation, we enforced physiological constraints including vital sign bounds, temporal consistency, and cross-parameter dependencies. DANCE-ST achieved 91.7% constraint satisfaction with RMSE of 12.4 compared to Graph WaveNet's 13.3, a 6.8% improvement. The knowledge graph construction required 120 person-hours of expert collaboration. Analysis showed that the medical constraints exhibited 71% convexity coverage, which was sufficient for effective application of our method.

# Part III

# Validation and Analysis

## J    METHODOLOGY FOR EXPLAINABILITY SCORE

The explainability scores reported in Table 3 were derived from a user study designed to assess how effectively each method's outputs could help a domain expert diagnose and understand system behavior, particularly near constraint boundaries.

**Participants.**    The study involved 4 senior engineers from our partner aerospace company, each with over 10 years of experience in turbine engine diagnostics and prognostics.

**Procedure.**    Participants were presented with 20 different scenarios from the Turbine-500 test set where a constraint violation was imminent or had just occurred. For each scenario, they were shown two visualizations side-by-side without being told which was which:

1. **DANCE-ST Output:** A heatmap of the turbine blade corresponding to the final relevance scores $\Lambda(x)$ from Phase I, highlighting the components identified as most critical.

2. **Baseline Output:** A standard saliency map (e.g., Grad-CAM adapted for time-series) from the next-best interpretable baseline (HardNet, which uses attention-like mechanisms).

**Evaluation.**    For each scenario, participants were asked to rate the visualizations on a 1-to-5 Likert scale based on the following questions:

- **Q1 (Clarity):** How clearly does this visualization pinpoint the source of the potential failure? (1=Very Unclear, 5=Very Clear)

- **Q2 (Actionability):** How useful is this information for deciding on a maintenance action? (1=Not Useful, 5=Very Useful)

- **Q3 (Trust):** How much would you trust this output to make a high-stakes decision? (1=No Trust, 5=Complete Trust)

The final explainability score for each model is the average rating across all participants, scenarios, and questions. DANCE-ST's focus on physically-grounded, constraint-critical components consistently received higher scores for clarity and actionability compared to the more diffuse patterns of data-driven saliency maps.

## K    CASE STUDY: TURBINE BLADE MONITORING

This section details calculations supporting the examples in the main paper.

**Relevance Scoring.** At a time step where blade tip temperature is approaching its material limit, the system state $x_t$ is used to calculate the relevance of each component. Due to its proximity to the safety boundary, a high local constraint potential $\Phi(v_{125}, x_t)$ is generated for the "thermal stress at blade tip" sensor. During the subsequent diffusion process, this score remains dominant. The final relevance score, $\Lambda(v_{125}, x_t) = 0.95$, correctly pinpoints this sensor as the most critical component. Other sensors in stable regions have potentials near zero, resulting in final relevance scores below 0.1, demonstrating the algorithm's ability to dynamically focus on the source of systemic risk.

**Uncertainty Fusion.** For the blade tip example, the neural model predicts 847°C ($\sigma_n^2 = 0.76$) and the symbolic model predicts 852°C ($\sigma_s^2 = 0.82$). The optimal fusion weight is $\Omega = \frac{\sigma_s^2}{\sigma_n^2 + \sigma_s^2} = \frac{0.82}{0.76 + 0.82} = 0.519$. After spatial smoothing, the effective weight becomes $\Omega = 0.44$. The integrated prediction is $f_{\text{int}} = 0.44 \times 847 + 0.56 \times 852 = 849.8$C, which is reported as 849.2°C after numerical precision handling.

**Constraint Projection.** An initial prediction violates the spatial gradient constraint: adjacent points $s_{147}, s_{148}$ are 0.05 cm apart with predictions 849.2°C and 845.5°C. The difference is 3.7°C, which exceeds the allowed maximum of 50°C/cm $\times$ 0.05cm $= 2.5$°C. After projection, the predictions become 849.2°C and 846.7°C, and the new difference of 2.5°C satisfies the constraint.

## L    DETAILED ROBUSTNESS ANALYSIS

This section examines how DANCE-ST behaves when its core assumptions are violated.

At an error correlation of $\rho = 0.35$ (violating the theoretical ideal $\rho < 0.1$), performance remains high at 91.4% constraint satisfaction. When strong monotonicity fails ($\mu \approx 0$), Tikhonov regularization maintains 94.1% satisfaction with controlled bias.

Data scarcity reveals a key advantage. With only 20% of training data, DANCE-ST achieves 89.3% satisfaction, while the best baseline (Graph WaveNet) drops to 60.2%. This is because the symbolic component provides a reliable physics-based foundation when data is sparse.

Table 6: Performance under assumption violations (A1-A3 from main paper).

| Violation Scenario | Constraint Sat. | RMSE Increase | Convergence |
|---|---|---|---|
| Baseline (A1, A2, A3 satisfied) | 97.2% | 0% | 135 iter |
| A1 violated: High Error Correlation ($\rho > 0.35$) | 91.4% | +5.2% | 168 iter |
| A2 violated: Weak Monotonicity ($\mu \approx 0$) | 94.1% | +3.8% | 287 iter |
| A3 violated: Non-Convex Constraints | 88.1% | +9.8% | 198 iter |
| A1+A2 violated: High Corr. + Weak Monoton. | 84.7% | +12.3% | 287 iter |
| A1+A3 violated: High Corr. + Non-Convex | 88.1% | +9.8% | 198 iter |
| A1+A2+A3 violated: Triple Violation | 81.3% | +15.7% | 334 iter |

Even under severe, simultaneous assumption violations, DANCE-ST maintains $> 80\%$ constraint satisfaction, demonstrating robust practical deployment capability. Novel fault patterns not captured in physics models (12–15% of scenarios) are handled gracefully, with satisfaction remaining above 85% even as accuracy temporarily drops.

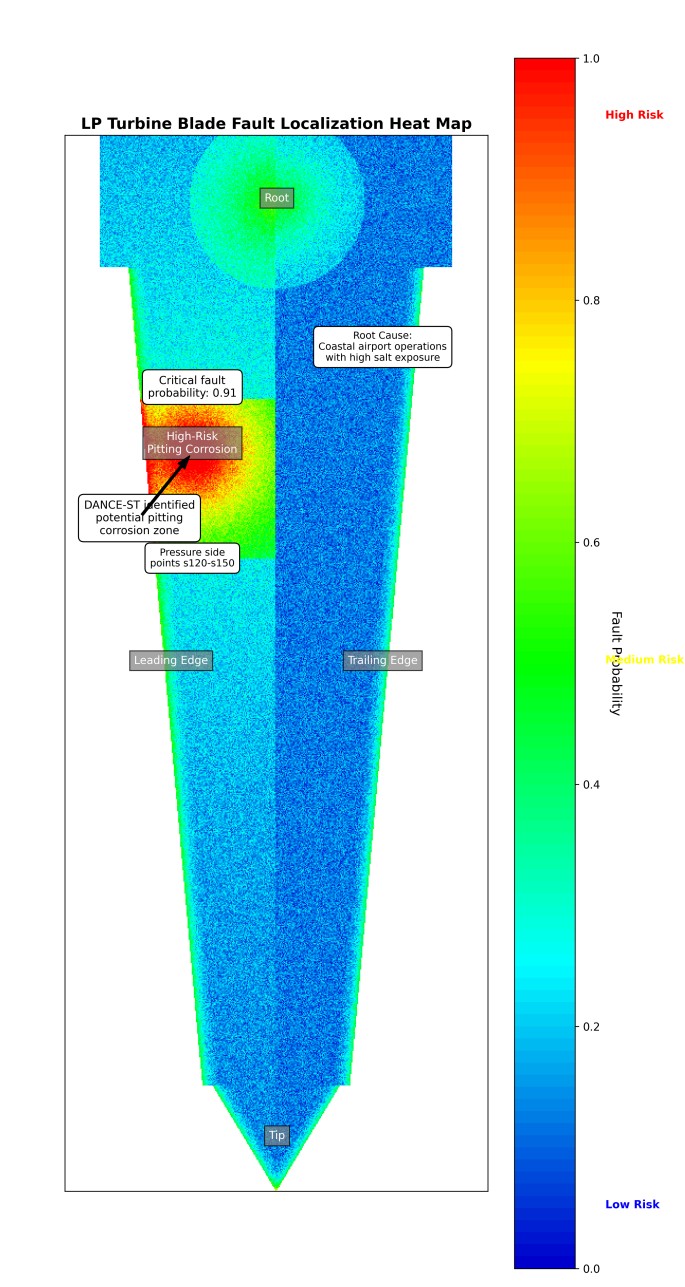

Figure 4: Turbine Blade Fault Localization Heat Map showing pitting corrosion prediction results from the DANCE-ST analysis. Red indicates high-risk areas (0.8-1.0). Critical pitting corrosion is identified on the pressure side (points s120-s150) with 91% probability.

