# OpenReview forum: "DANCE-ST: Why Trustworthy AI Needs Constraint Guidance, Not Constraint Penalties"
_ICLR.cc/2026/Conference — ICLR 2026 Conference Desk Rejected Submission_

### Official Review · Reviewer_QQrS · 2025-10-18

**Soundness:** 1
**Presentation:** 1
**Contribution:** 1
**Rating:** 2
**Confidence:** 4

**Summary:**

The paper proposes a framework for enforcing output constraints during the deployment of forecasting models for physical systems. The method combines a pre-trained neural network with a separate physics-based symbolic model. The system is represented as a graph where vertices correspond to components to be forecasted and edges encode their physical dependencies.

The algorithm proceeds in three phases. First, it identifies the top-k components that are most at risk of violating their constraints at the current time t. For these selected nodes, the prediction for t+1 is computed as a weighted average of the neural and symbolic models, with weights shifting toward the symbolic model when the system is close to violating a constraint. The paper does not specify how predictions are obtained for the remaining, stable nodes. Finally, the full prediction vector for all components is projected onto the feasible set to guarantee physical validity of the output.

**Strengths:**

**Originality**. The paper’s main originality lies in its architectural design, which integrates three distinct ideas: relevance-based node selection, neurosymbolic fusion, and a projection step. The selective focus on a subset of nodes also provides built-in interpretability.

**Quality**. The paper presents an extensive empirical evaluation across three industrial datasets, supported by a detailed ablation study and robustness analysis.

**Clarity**. There is not much to praise in terms of clarity.

**Significance**. The work addresses a highly relevant problem—ensuring safety in AI systems deployed in physical environments. The emphasis on interpretability is a welcome addition.

**Weaknesses:**

The paper has numerous weaknesses which underscore my recommendation for rejection.

### Flawed Experimental Section

1. **HardNet Baseline is Fundamentally Misrepresented**.
The paper's experimental validation appears to be flawed due to two major inconsistencies in the HardNet baseline results.

    First, the paper reports a *~2% constraint violation rate for HardNet*. This is inconsistent with the method’s design; HardNet computes outputs via a closed-form projection that guarantees feasibility, and the original HardNet paper reports *0 constraint violations* across all experiments. A deviation of this magnitude cannot reasonably be attributed to minor numerical drift and instead strongly suggests an implementation error.

    Second, the reported *>40 second inference time for HardNet is unlikely*. HardNet’s projection is closed-form, and the original paper reports inference in the *millisecond range* for a problem of comparable scale (100 variables and 100 constraints). While the hardware differs, this alone cannot account for a discrepancy of almost four orders of magnitude. This reinforces the suspicion that the authors relied on an incorrect or at the very least severely unoptimized implementation of a key baseline, which *undermines the credibility of the entire experimental comparison*.

2. **The Method Fails to Guarantee Feasibility**.
The proposed method fails to meet the primary requirement of a framework for enforcing constraints: guaranteeing feasible solutions. The paper reports only 97.2% constraint satisfaction, a critical failure for a method intended for safety-critical systems where even rare violations can be catastrophic.

3. **The Paper's Efficiency Claims are Misleading**
The paper's claims of efficiency are misleading, as the reported speed is achieved at the direct cost of feasibility. A rigorous evaluation would have analyzed the trade-off between more iterations and achieving a 100% guarantee—a trade-off the authors fail to address.

    Furthermore, any claims of practical efficiency are undermined by the high upfront cost of constructing the knowledge graph, which the authors state requires 120 person-hours per domain.

### The Framework's Complexity is Unjustified
4. **The Framework's Complexity is Unjustified**.
At its core, the proposed framework is a projection method with additional, complex pre-processing stages, but the necessity of these extra steps is not convincingly justified. The authors argue that this pre-processing improves final performance, citing an ablation study where removing the relevance-selection filter (Phase 1) causes a catastrophic drop in performance.

    However, their interpretation of this result is questionable. The authors claim the performance drop occurs because the model "loses focus on the most relevant signals," but they fail to disprove an alternative hypothesis: that the subsequent fusion and projection stages are fragile, and that Phase I's primary function is merely to shield them from the full problem's complexity by reducing the input size. The lack of a proper control experiment (e.g., applying Phases 2 and 3 on a random subset of nodes) leaves this justification unsubstantiated.

    The ablation study shows that removing Phase I also causes a significant 3.0 percentage point drop in constraint satisfaction. The paper offers no explanation for why that would also harm feasibility.

### The Theoretical Foundations are Unsound
5. **The Paper Lacks a Universal Approximation Guarantee**
A major theoretical flaw is the absence of a universal approximation guarantee. This is required to show that an architecture modified to enforce constraints can still represent the optimal feasible solution.

6. **Unsound Proof of Lemma C.1**
The proof of Lemma C.1—which underpins the robustness claim in Theorem C.1—is mathematically unsound. Its final step asserts that for$ \Omega, \alpha' \in [0,1] $, the difference $ |\Omega - \alpha'| \le \tfrac{1}{2} $ must hold. This bound is incorrect, and no argument is provided to justify why the computed fusion weight $ \Omega $ should necessarily lie within 0.5 of the “true” interpolation factor $ \alpha' $.

If the bound is relaxed to the correct range $ |\Omega - \alpha'| \le 1 $, the result becomes vacuous—it merely states that the fusion error cannot exceed the total disagreement between predictors, which offers no nontrivial robustness guarantee. Consequently, the claim of a *provably robust* fusion process is unsupported unless a missing justification establishes the tighter 1/2 bound.

### Poor scientific practices

7. **Overclaiming and Lack of Scientific Rigor**.
The paper fails to properly scope its contribution and makes several claims that are not supported by evidence.

    The authors overclaim the significance of their work, framing a method for physics-informed forecasting as a "foundational paradigm for all trustworthy AI". This is an overstatement, as the framework relies on several strong assumptions—such as the existence of a graph structure, a valid physics-based model, and predominantly convex constraints—that are not generalizable to many other domains.

    Other examples of misleading claims appear in Section 3.3. For instance, the paper introduces an error correlation bound of $\rho < 0.35$ between the neural and symbolic models and a convexity parameter of $\mu > 0.03$, presenting them as if they were general principles rather than quantities tied to a very specific and narrow problem setting, which is never clearly specified.

8. **Methodological Gaps Make the Work Irreproducible**
The authors state that their core logic is applied only to the top-k "risky" nodes identified in phase 1 but never specify what prediction is used for the remaining, stable nodes. However, the final projection in Phase 3 requires a full system vector to ensure global consistency across coupled constraints.

9. **Poor Presentation**
The paper's presentation is not up to the standards of a scientific publication. The writing is often unclear, and the core methodology is difficult to follow. The use of informal language (e.g., "hasn't") further contributes to a general lack of rigor.

**Questions:**

1. The authors do not report any LLM use. Can they confirm that they did not use LLMs in the development of their work?

2. What prediction model is used for the remaining nodes not selected by the top-k filter in Phase I?

3. Why did you not consider a hybrid residual model, where a neural network predicts the symbolic model’s error and the final output is then projected? This seems like a more straightforward approach, as it avoids the need to tune weights for the fusion of the two models. Moreover, because this formulation relies heavily on the symbolic model by design, it should be more naturally inclined toward feasibility, potentially making the projection step easier or more successful in terms of performance.

---

### Official Review · Reviewer_4uXA · 2025-10-27

**Soundness:** 3
**Presentation:** 3
**Contribution:** 3
**Rating:** 6
**Confidence:** 3

**Summary:**

This paper proposes DANCE-ST, a novel constraint-guided learning framework for trustworthy spatiotemporal prediction. Instead of treating physical constraints as loss penalties or post-hoc enforcement, DANCE-ST interprets them as collaborative information sources that actively guide the learning process.
The method proceeds through three phases:

1. Constraint-Potential Diffusion identifies critical system components by propagating state-dependent “constraint potentials” over a knowledge graph.

2. Neurosymbolic Fusion combines neural and physics-based predictions with variance-based weighting and theoretical error bounds for asynchronous sensors.

3. Structure-Exploiting Projection ensures final predictions satisfy physical constraints via an adapted Douglas–Rachford scheme with guaranteed linear convergence.

The entire pipeline operates within a fault-tolerant multi-agent architecture, providing robustness to component failures. Experiments on multiple industrial datasets (C-MAPSS, Turbine-500, FEMTO Bearings) and a medical dataset (MIMIC-III) demonstrate high constraint satisfaction (97.2%), strong predictive accuracy, and superior interpretability (4.6/5 explainability score).

**Strengths:**

The paper presents a highly original conceptual shift in the field of trustworthy AI and physics-informed learning. Instead of viewing physical or safety constraints as penalties imposed after prediction, DANCE-ST treats them as collaborative guides that actively shape learning throughout the pipeline.

In terms of quality, the work demonstrates exceptional technical depth. Each phase of the DANCE-ST framework is rigorously developed, theoretically grounded, and empirically validated. The authors provide mathematical convergence guarantees, uncertainty-aware fusion bounds, and extensive ablation studies, which collectively demonstrate the robustness and reliability of the approach.
Regarding clarity, the paper is generally well organized and methodically presented.

Finally, the significance of the work is high. The approach bridges gaps between deep learning, physics-based modeling, and explainability. It provides both theoretical insights (e.g., constraint-guided optimization geometries) and practical solutions (fault-tolerant deployment, interpretability metrics) that can influence future research in safety-critical machine learning.

**Weaknesses:**

First, the framework relies heavily on manually constructed knowledge graphs, requiring around 120 person-hours per domain. This dependence on expert modeling limits scalability and reproducibility.

Second, while the paper’s central idea is conceptually original, the individual technical components (graph diffusion, uncertainty-weighted fusion, Douglas–Rachford projection) are adapted from established methods. The contribution therefore lies in integration rather than algorithmic novelty.

Third, the theoretical guarantees rely on strong convexity and monotonicity assumptions that may not hold in complex, real-world systems with non-convex constraints.

Finally, the interpretability evaluation lacks methodological transparency (the reported 4.6/5 expert score is not supported by details on rating criteria or sample size). Providing clearer evaluation protocols or qualitative case studies would substantiate these claims.

**Questions:**

1. Clarifying what is theoretically new in the fusion mechanisms (this DANCE-ST algorithm), or providing additional ablations comparing with standard versions, would better highlight genuine methodological advances.

2. Theoretical guarantees rely on convex constraints, but many real systems involve non-convex or discontinuous ones. How does DANCE-ST perform when these assumptions are violated, and could the authors provide empirical evidence or discussion on robustness under non-convex conditions?

3. The reported 4.6/5 interpretability score is compelling but under-detailed. How many experts participated, what criteria were used, and was inter-rater agreement assessed?

---

### Official Review · Reviewer_YExx · 2025-10-29

**Soundness:** 2
**Presentation:** 1
**Contribution:** 2
**Rating:** 2
**Confidence:** 4

**Summary:**

This paper presents DANCE-ST (Distributed Agent Network for Constraint-Enabled Spatiotemporal prediction). DANCE-ST is concerned with improved predictions where, e.g., physical constraints such as material properties need to be respected. To this end, a neural predictor $f_n$ and a symbolic predictor $f_s$ are employed and integrated into a combined predictor $f_{int}$. The authors demonstrate on three datasets (NASA C-MAPSS, Turbine-500, and FEMTO Bearing) competitive accuracy while improving on constraint satisfaction, computation time, and explainability scores.

**Strengths:**

- The related work section provides a good overview of competing approaches and Tables 1 & 2 especially help understanding how DANCE-ST is supposed to differ.
- The integration of neural and symbolic prediction described in Sections 3 and 4 is intuitive and easy to follow.
- The ablation study shows well how each phase of DANCE-ST provides an important benefit to the task.

**Weaknesses:**

- Sections 1 and 2 provide numbers and claims about DANCE-ST's performance that lack context to fully understand their meaning or impact. For example, Table 1 shows a checkmark for DANCE-ST on all columns with added annotations such as (38.4s) for real-time capabilities or (multi-agent) for scalability. Without context, such as the competing methods' runtime (on the same problem and hardware), the the two sections felt uninformative and puzzling.
- The paper often leaves me puzzled on how some concrete numbers are justified (DANCE-ST converges in 135 iterations, "generic methods" require > 1000 iterations, ...). For example, is there a citation for the specific correlation boundary ($\rho < 0.35$) on the discussed failure modes in line 197?
- The paper states that "While HardNet achieves the highest constraint satisfaction (98.1%), DANCE-ST delivers competitive constraint satisfaction (97.2%) with superior performance in accuracy, efficiency, and interpretability". But, Table 3 does not show superior accuracy, with numbers being extremely similar between HardNet and DANCE-ST.

**Questions:**

I have no questions to the authors.

---

### Official Review · Reviewer_L4hc · 2025-10-31

**Soundness:** 2
**Presentation:** 2
**Contribution:** 3
**Rating:** 4
**Confidence:** 2

**Summary:**

The paper proposes DANCE-ST, a three-phase, constraint-guided framework (relevance selection →  neuro-symbolic fusion →  structure-aware projection) for spatiotemporal prediction under physical constraints. It also targets high accuracy and high constraint satisfaction, and uses multi-agent deployment.

**Strengths:**

- The idea of using constraints as guides rather than penalties is fresh.
- The paper presents strong quantitative results and a well-organized table with plenty of comparisons.
- The paper is pleasant to read.

**Weaknesses:**

- Phase I proposes an appealing idea of constraint-potential diffusion to identify critical components, but it relies on overly strong assumptions. The method assumes that the physical coupling graph is accurate, static, and sparse, whereas real-world systems are often nonlinear, time-varying, and uncertain.

- The diffusion mechanism is linear and heuristic, rather than physically derived, which limits the realism of the proposed “guidance.”

- The reported interpretability score (4.6 / 5) lacks details to fully understand and appreciate it.

- The paper would be better with stronger ablation studies, especially with different hyperparameter settings.

- There is also a citation issue: the reference “Jianwei Zheng et al., ICLR 2025” lists the wrong authors. The correct authors are Ricardo Buitrago Ruiz, Tanya Marwah, Albert G and Andrej Risteski. Please verify and correct all references.

- The writing is vague in several places. Many abbreviations are not properly introduced (e.g., QP, MCP introduced in the appendix, A2A), and important concepts such as the knowledge graph appear without background. This makes it hard for reviewers unfamiliar with the terminology to follow the logic of your narrative.

- Figure 1 is dense and difficult to interpret due to many unexplained acronyms.

**Questions:**

1. For the interpretability score, please clarify the number of raters, domains evaluated, and example tasks.

2. What are the effects of hyperparameter variation on the results?

---

### Note · Program_Chairs · 2025-12-10
**Submission Desk Rejected by Program Chairs**

Hallucinated reference: Sardar Asif, Saad Ghayas, Waqar Ahmad, and Faisal Aadil. Atcn: an attention-based temporal convolutional network for remaining useful life prediction. The Journal of Supercomputing,